# Soft and Hard Total Quality Management Practices Promote Industry 4.0 Readiness: A SEM-Neural Network Approach

Kashif Ali [1], Satirenjit Kaur Johl [1,*], Amgad Muneer [2,3], Ayed Alwadain [4] and Rao Faizan Ali [5,*]

1 Department of Management Sciences, Universiti Teknologi Petronas, Seri Iskandar 32160, Malaysia
2 Department of Computer and Information Sciences, Universiti Teknologi Petronas, Seri Iskandar 32160, Malaysia
3 Department of Imaging Physics, The University of Texas MD Anderson Cancer Center, Houston, TX 77030, USA
4 Computer Science Department, Community College, King Saud University, Riyadh 145111, Saudi Arabia
5 Department of Software Engineering, University of Management and Technology, Lahore 54728, Pakistan
* Correspondence: satire@utp.edu.my (S.K.J.); faizan.ali@umt.edu.pk (R.F.A.)

**Abstract:** Industry 4.0 (I4.0) is a technological development in the manufacturing industry that has revolutionized Total Quality Management (TQM) practices. There has been scant empirical research on the multidimensional perspective of TQM. Thus, this study aims to empirically examine the effect of the multidimensional view of TQM (soft and hard) on I4.0 readiness in small and medium-sized (SMEs) manufacturing firms. Based on the sociotechnical systems (STS) theory, a framework has been developed and validated empirically through an online survey of 209 Malaysian SMEs manufacturing firms. Unlike the existing TQM studies that used structural equation modeling (SEM), a two-stage analysis was performed in this study. First, the SEM approach was used to determine which variable significantly affects I4.0 readiness. Second, the artificial neural network (ANN) technique was adopted to rank the relative influence of significant predictors obtained from SEM. The results show that the soft and hard TQM practices have supported the I4.0 readiness. Moreover, the results highlight that hard TQM practices have mediating role between soft TQM practices and I4.0 readiness. The ANN results affirmed that customer focus is considered an important TQM factor for I4.0 managerial readiness, advanced manufacturing technology for operational readiness and top management commitment for technology readiness. In a nutshell, the SEM-ANN approach uniquely contributes to the TQM and I4.0 literature. Finally, the findings can help managers to prioritize firms' soft and hard quality practices that promote I4.0 implementation, especially in emerging economies.

**Keywords:** Total Quality Management; soft TQM; hard TQM; Industry 4.0; readiness

## 1. Introduction

In today's disruptive business environment, the survival of small and medium enterprises (SMEs) is crucial for both developed and emerging economies. Manufacturing SMEs are considered the economy's driving force in industrial economies. As the backbone of the economy, the role of SMEs has gained significant attention in the scholarly world, especially in the age of digital transformation. The term digital transformation is generally known as Industry 4.0 (I4.0) or the fourth industrial revolution [1]. Piccarozzi, et al. [2] define I4.0 is the integration of digital technologies with firm value chain that enable to deliver flexible and adaptable business structures in order to cope with a changing business environment. According to Machado, et al. [3], digital transformation or I4.0 is the strategic transformation of all business operations, creating a new ecosystem where technology generates and transfers value to the stakeholders, enabling the organization to respond more swiftly to changing circumstances. Gerald C. Kane, et al. [4] argued that I4.0 is not technology-centric but includes people and processes. Likewise, Dias, et al. [5] claimed that the application of I4.0 transforms all organizational areas such as people, processes, and technology. In other

words, the survival of a firm depends on how social (people) and technical (process and technology) interact with each other to gain a competitive advantage [6].

In the I4.0 literature, the readiness term is used to understand the firm ability to take advantage of I4.0 technologies. In other words, I4.0 readiness highlight how an organization is being digitally prepared for I4.0 technologies [7]. In line with I4.0, manufacturing firms are under pressure to transform from a labor-intensive to a digitalization landscape [8]. Although I4.0 has several promising advantages, many firms, especially SMEs, struggle with its implementation [9]. For instance, SMEs manufacturing in West Virginia, USA, still struggle to adopt I4.0 technologies [10]. Stentoft, et al. [11] argued that European SMEs have a low level of I4.0 acceptance. According to OECD [12] report, organizations that fall under the SMEs category have a low level of readiness for I4.0. In the emerging economies context, the Malaysian Economic Planning Unit [13] highlighted that SMEs are three times less productive than large firms. The leading cause of this was the low level of I4.0 readiness.

The existing literature suggests that SMEs are not ready to implement I4.0. For example, Khin and Kee [8] argued that digital transformation is not an easy task; it requires both social and technical aspects to move out of their conventional practices. Furthermore, it also requires new expertise, resources, and commitments to a greater extent [14]. Therefore, sound preparation and readiness factors are essential to I4.0 successful implementation [15,16]. Although several I4.0 readiness assessment frameworks are available in the extant literature, the proposed I4.0 readiness frameworks do not differentiate between SMEs and large organizations [9]. Recently, Khin and Kee [8] developed an I4.0 readiness framework for SMEs in emerging economies which is divided into managerial, operational, and technology readiness.

Furthermore, the growing body of literature recognizes the importance of appropriate strategies to promote I4.0 initiatives. For instance, Crešnar, et al. [17] affirmed that Total Quality Management (TQM) is an important strategy to promote I4.0 readiness in manufacturing. Stentoft et al. [11] highlighted that firms' social and technical factors are essential to promote I4.0 readiness and actual practices in manufacturing SMEs. Many qualitative and review studies suggested that effective TQM implementation help firms improve I4.0 implementations [16–18]. Additionally, the practical studies highlighted that TQM is still widely used as a strategy.

According to management tools and trend survey report of Bain and Company, it is highlighted that more than 75% of firms are satisfied with TQM practices, and it remained in the top 25 management tools from 1993 to 2017 [19,20]. This report further affirmed that the TQM approach earned the highest satisfaction rate (4.09) among other management tools [19]. However, little attention has been paid to TQM practices to promote I4.0 readiness and actual practices, especially in manufacturing SMEs [8,11]. Moreover, the previous research studies on TQM-I4.0 are based on qualitative research [21,22], case study [6] and review [1,5]. Although a few empirical studies are available, they focus on factors identification [16,23] and employee perception [24]. Therefore, knowledge and practice gaps exist, and this research tries to fill these gaps. Based on these issues and the literature gaps, this study will fill these gaps by addressing the following research objectives.

RO1: To analyze the association between social and technical TQM practices to achieve I4.0 readiness in manufacturing SMEs.

RO2: To examine the importance of social and technical TQM practices to promote I4.0 readiness.

The remainder of this paper is organized as follows. The next section presents the literature review of this study, followed by a theoretical framework and hypothesis development. Then, the next section presents data analysis, followed by a discussion of the findings. Finally, the last section highlights research contributions, limitations, and future work.

## 2. Literature Review

### 2.1. Industry 4.0 Readiness

In the past literature, I4.0 readiness has been described as the degree to which an organization can take advantage of digital technologies [7]. According to Črešnar, Potočan and Nedelko [17], I4.0 readiness refers to the organization's willingness to incorporate and implement I4.0 practices in social and technical practices. The prior literature highlights that I4.0 readiness frameworks was developed with two unique perspectives: identifying the users of I4.0 and determining the practical application of readiness models [7]. However, it has been observed that I4.0 readiness models developed at academic levels are broadly not recognized in industrial world. The review work of Hizam-Hanafiah, Soomro and Abdullah [7] argued that academic I4.0 readiness models are incompatible with the fast-moving industry. Moreover, the purpose of the I4.0 readiness model development is different, as certain models are designed for specific industries and sectors, and some have a narrow scope. Additionally, the purpose of I4.0 readiness models can vary in terms of short, moderate, and long-term. However, a great deal of work has been put into developing models of I4.0 readiness, and there is a lack of disagreement regarding the effectiveness of these models. However, extensive models have been developed on I4.0 readiness, and lack of disagreement on their efficiency. However, these models are a management tool for reconfiguration, realignment, and renewal of organizational existing resources and capabilities. Moreover, Felch, et al. [25] argued that it is critical to study existing I4.0 readiness models, whether they originated from the practical or scientific community. Thus, Table 1 shows the existing I4.0 readiness models.

**Table 1.** Past Industry 4.0 models.

| Academic/Industry | Year | Model Name | References |
|---|---|---|---|
| Academic | 2022 | I4.0 readiness of manufacturing SMEs | Khin and Kee [8] |
| | 2021 | I4.0 readiness of technology companies | Soomro, et al. [26] |
| | 2018 | I4.0 business model innovations tools | Müller and Voigt [27] |
| | 2018 | I4.0 adoption model for manufacturing firms | Mittal, Khan, Romero and Wuest [10] |
| | 2017 | I4.0 readiness model for tool management | Schaupp, et al. [28] |
| | 2016 | Design business modeling for I4.0 | Gerlitz [29] |
| | 2017 | Reference architecture model for I4.0 (RAMI4.0) | Kannan, et al. [30] |
| | 2006 | I4.0 readiness model for manufacturing | Banthita and Salinee [31] |
| Industry | 2018 | Benchmarking readiness I4.0 | [7] |
| | 2016 | I4.0 introduction strategy | |
| | 2014 | I4.0 barometer | |
| | 2014 | Roland Berger I4.0 readiness index | |

Based on the past literature, this study is based on three-dimensional I4.0 readiness framework. According to Khin and Kee [8], I4.0 readiness framework consist of managerial, operational, and technological readiness. Managerial readiness refers to top management commitment and priority towards digital transformation. The operational readiness means that the preparation in term of human resources, finance, and infrastructure. Finally, the technological readiness indicates the preparation in the form of technical skills and knowledge of the workforces and the digital readiness of a system [8].

### 2.2. Industry 4.0 Readiness in Malaysia

Based on World Bank [32] report, Malaysia falls under the upper-middle countries group and is classified as an emerging economy. Although Malaysia has abundant natural resources, the manufacturing sector rapidly turned the country into an industrialized nation [33]. The Malaysian Economic Planning Unit [13] report highlighted that the manu-

facturing sector contributed 22.9% of the country's GDP during 2019–2020. Although the COVID-19 pandemic affected the economic growth, the manufacturing sector performed better than other sectors. In the I4.0 context, the manufacturing sector experienced stiffening pressures to adopt digital technologies. To promote I4.0, the Malaysian government launched a framework, 'Industry4WRD' in 2018. The 12th Malaysian plan (2021–2025) highlighted that the Malaysian manufacturing sector faced severe challenges at the global level. Moreover, Malaysia has moved away from being an investment destination because of the low adoption of technologies and labor productivity. The Malaysian Productivity Corporation [34] report highlighted that productivity in SMEs remains low, despite various efforts undertaken to boost it. Khin and Kee [8] argued that I4.0 readiness remains low in Malaysian manufacturing SMEs because it requires massive investment, and SMEs are unsure where to start. From the literature point of view, Soomro, Hizam-Hanafiah, Abdullah, Ali, and Jusoh [26] conducted a pilot study to examine the I4.0 readiness in Malaysian technology companies. The findings affirmed that 69% of firms feel pressure to work in I4.0, 84% are willing to take risks, 82% of technology firms understand I4.0, and 90% have top management support. The qualitative study of Khin and Kee [8] highlighted that lack of training, skills, and technical resources are impeding factors in manufacturing firms. Hizam-Hanafiah, Soomro and Abdullah [7] argued that large and medium-sized firms are in a better position to leverage various I4.0 technologies compared to small industries.

*2.3. TQM and I4.0 Readiness*

In the new global economy, social and technical factors promote I4.0. To promote I4.0 implementation in an organization, the literature suggested that a holistic and integrated approach is appropriate that encompasses both social and technical factors [1,18,23]. According to Manz and Stewart [35], TQM is an integrated approach that consists of both social and technical systems to achieve firm stability, facilitating short- and long-term success. In TQM studies, social factors represent soft TQM practices, and technical factors represent hard TQM practices [1]. Moreover, researchers argued that TQM in I4.0 can be treated as integrating social and technical factors [36]. The past studies highlighted that TQM practices align with I4.0, which helps organizations promote efficiency, performance, and improved business models [37,38]. In the contemporary literature, soft and hard TQM practices are essential for implementing I4.0 practices in SMEs [8].

To implement I4.0 at the firm level, the existing literature highlight the various factors are responsible such as top management [21,39], customer focus [37,40], training and learning [41], HRM/teamwork [42], process management [43] and quality information and analysis [41,44]. Antony, Sony, Furterer, McDermott, and Pepper [36] argued that top management provides a strong foundation of values and policies and essential resources to implement I4.0 readiness. Similarly, Chiarini and Kumar [39] argued that top management commitment and involvement are vital to implementing I4.0. Moreover, in terms of customer perspective, customer demands are more dynamic in the current competitive milieu. They require a product of good quality at a reasonable price in a shorter period [40]. Thus, if organizations are more focused on customer demands, they give more importance to I4.0 technology to satisfy them [45]. The successful execution and implementation of I4.0 require good quality training because it promotes critical thinking and social skills essential for I4.0 implementation [41]. Further, in the current digital environment, human resource is vital to implementing I4.0 because human resources with teamwork skills can perform multiple responsibilities and work more systematically with digital devices [46]. In addition, Ali and Johl [1] highlight that process management, as a hard TQM, is vital for digitalization. In I4.0, the personalized production process can help meet increasing customer demands and needs [37]. Moreover, the manufacturing process's automation could help improve conformance quality [38]. Finally, the effective implementation of I4.0 is based on quality-related data collection, evaluation, and decision-making [38].

Overall, TQM as an integrated approach (soft and hard) becomes essential in implementing I4.0 in manufacturing firms. Moreover, Črešnar, Potočan and Nedelko [17]

concluded that TQM practices ascertain a vital building block for I4.0 readiness in manufacturing firms.

### 2.4. Multidimesnional View of TQM

According to the literature, Wilkinson [47] was probably the first to categorize TQM into hard and soft TQM. The hard TQM involves production techniques such as product design, processes, and procedures, whereas soft TQM represents a social system encompassing human resources management and establishing customer awareness [48]. Based on this classification, this study draws on prior research studies' work [49–51] to construct the soft and hard TQM.

In the prior literature, researchers have tried to differentiate between the soft and hard aspects of TQM. The soft TQM aspects relate to behavioral characteristics and generally deal with human resources, social, and organizational cultures [48,51]. On the other hand, hard TQM focus on technical aspects, referring to management tools, techniques, and practices [23] (Babatunde, 2020). Over the decades, researchers extended the general perspective on TQM critical factors, but there is a disagreement among quality researchers [52]. Similar to the study of Ali and Johl [1,53] this study considers the critical success factors for soft TQM (top management commitment (TMC), customer focus (CF), training and learning (EDT), and hard TQM (Process management (PM) and Quality information and analysis (QIA).

In applying the STS perspective in TQM, quality researchers have maintained the co-existence of social and technical systems in quality management. Zeng, Zhang, Matsui, and Zhao [48] argue that the quality management system collectively encompasses a sociotechnical mix of tactics. Human aspects such as TMC, CF, EDT, and HR represent the socio-end of the continuum of design whereas the PM and QIA are at the opposite technical end of the continuum. Furthermore, Sony and Naik [54] stated that for the effective implementation of I4.0, STS is an integral theoretical component.

## 3. Theoretical Framework and Hypothesis Development

### 3.1. Sociotechnical Systems (STS) Theory

Sociotechnical systems theory is often used to explain the dynamic interplay between people and technology. Neither people nor technology can be considered in isolation to maximize performance [55]. Sociotechnical systems theory is based on two factors or systems: socio (society and people) and technical (machines and technology) [56]. A group of researchers originally developed it at the Tavistock institution in London in the 1950s. Then, Trist and Bamforth [57] presented the philosophical and epistemological work. They argued that although technological changes are pretty rational, if firms ignore employee needs, it may reduce the benefits expected from new technology [55].

According to STS theory, organizations are made up of two independents but connected systems: a social system and a technical one. In comparison, the social systems focus on the relationships among people and their attributes, such as values, skills, and attitudes. In contrast, the technical system focuses on the tasks, processes, and technologies to produce designated output [48]. STS theory is founded on two primary principles: interaction of social and technical factors and joint optimization between social and technical considerations rather than emphasizing one over the other [55,58]. According to Sony and Naik [54], I4.0 is a sociotechnical system. For the sustainable and successful implementation of I4.0, the joint optimization of social and technical factors is necessary [21,59,60].

In the extant literature, the STS theory has been adopted by Sciarelli, Gheith, and Tani [51] to empirically examine the soft–hard quality management effect on innovation and organizational performance. Babatunde [23] maps the competencies and implications for I4.0 to the soft and hard TQM perspective. Another study was conducted by Zeng, Zhang, Matsui, and Zhao [48] to investigate the impact of organizational context on soft–hard QM and innovation performance. Alkhaldi and Abdallah [61] developed a framework on STS theory to analyze the effect of soft and hard TQM practices on quality performance.

Marcon et al. [62] established a framework on STS theory to examine the organizational factors (social and technical) to achieve I4.0 adoption levels. Recently, Nguyen, Tucek, and Pham [21] argued that traditional TQM practices are based on standardization while I4.0 focuses more on technical. Hence, the role of people in I4.0 seems to be muted. Therefore, the STS theory is best suited to resolve this issue.

*3.2. Hypotehses Development*

3.2.1. Relationship between Soft and Hard TQM Practices

Based on STS theory, a successful system is the outcome of the synchronized alignment of social and technical systems [62], which articulates the relationship between soft and hard TQM practices [62]. Although hard TQM factors are essential, simply focusing on them may not increase competitive advantage on the long-term [48] because competitors will easily imitate and adopt these factors. In contrast, soft TQM practices are people-related factors that are not easily reproduced [51]. Thus, soft and hard TQM practices are essential to gain long-term competitive advantages [1,43,53]. Moreover, the TQM practices in terms of soft and hard support the idea of STS theory that both systems must be integrated and optimized to achieve higher performance. Tarí, et al. [63] argued that a firm's focus on customers would facilitate data collection about customer needs, which can be incorporated into the process to achieve better results. Similarly, employee training and learning can stimulate effective data collection and analysis [51]. Therefore, soft factors support hard TQM factors to achieve superior outcomes.

Moreover, the prior literature highlighted that a successful organization could understand the role of soft aspects in promoting hard TQM practices. For instance, Calvo-Mora, et al. [62] found that leadership and people management (soft) constitute the essential basis for process and strategy (hard) from a sociotechnical perspective to achieve higher performance. The study of Sciarelli, Gheith, and Tani [51] affirmed that soft TQM practices have a positive association with hard TQM practices. Khan and Naeem [64] depicted that soft TQM dimensions positively impact hard TQM dimensions. Tarí, Claver-Cortés, and García-Fernández [63] highlighted that soft practices promote hard TQM practices in manufacturing firms. Finally, Nasaj and Al Marri [65] asserted that soft factors significantly predict hard factors. Therefore, the following hypotheses have been proposed based on STS's theoretical basis and past empirical support.

**H1.** *Top management commitment (TMC) has positive relationship with hard TQM practices.*

**H1a.** *TMC has a positive Relationship with process management (PM).*

**H1b.** *TMC has a positive Relationship with quality information and analysis (QIA).*

**H1c.** *TMC has a positive Relationship with advanced manufacturing technology (AMT).*

**H2.** *Customer focus (CF) positively affects TQM practices.*

**H2a.** *CF has a positive Relationship with PM.*

**H2b.** *CF has a positive Relationship with QIA.*

**H2c.** *CF has a positive Relationship with AMT.*

**H3.** *Employee training and learning (EDT) has positive affect on hard TQM practices.*

**H3a.** *EDT has a positive Relationship with PM.*

**H3b.** *EDT has a positive Relationship with QIA.*

**H3c.** *EDT has a positive Relationship with AMT.*

3.2.2. Mediating Role of Hard TQM Practices

From the STS theory perspective, soft and hard TQM practices are two independent but interconnected factors. In the prior literature, soft TQM practices affect organiza-

tional performance via hard TQM practices [50]. Calvo-Mora, Blanco-Oliver, Roldán, and Periáñez-Cristóbal [66] examined the mediating role of hard TQM between soft TQM and outcomes variables. Similarly, Zeng, Zhang, Matsui, and Zhao [48] analyzed the mediating role of hard TQM between soft TQM and innovation performance in manufacturing firms. Calvo-Mora, et al. [67] examined the mediating role of TQM technical (hard) factors between social TQM practices and performance outcomes (customer, people, and society). The structural model results confirmed the mediating role of hard TQM practices in Spanish manufacturing industries. Khan and Naeem [64] claimed that hard quality practices mediate between soft quality practices and services innovation and manufacturing firm performance in developing countries. Moreover, Gambi, et al. [68] analyzed the mediating role of hard TQM between soft quality aspects and outcomes variables. Nasaj and Al Marri [65] examined the mediating role of hard quality factors between soft quality factors and performance outcomes. The following hypothesis has been proposed based on the literature and theoretical support.

**H4.** *The relationship between soft TQM (TMC, CF and EDT) practices and I4.0 readiness (MR, OR, and TR) is mediated by hard TQM (PM, QIA, and AMT) practices.*

**H4a.** *The relationship between soft TQM practices and I4.0 readiness (MR, OR, and TR) is mediated by PM.*

**H4b.** *The relationship between soft TQM practices and I4.0 readiness (MR, OR, and TR) is mediated by QIA.*

**H4c.** *The relationship between soft TQM practices and I4.0 readiness (MR, OR, and TR) is mediated by AMT.*

3.2.3. Relationship between TQM Practices and I4.0 Readiness

In today's business environment, TQM is an enabler of business excellence in I4.0. From the STS theory perspective, Sony and Naik [54] stated that socio (TMC, employee's education and training) and technical (automated data management and process management) are the critical enablers factors for I4.0 readiness. The exploratory study of Babatunde [23] mapped the I4.0 implications and competencies to soft and hard TQM. The outcomes depicted the importance of soft and hard TQM practices for I4.0 from the STS theory perspective.

Thekkoote [41] argued that a strong top management commitment supports resource allocation and distribution and motivates employees to use and accept quality practices in I4.0. Similarly, Sureshchandar [40] acknowledged the role of leadership/top management in the effective implementation of I4.0. Dubey and Gunasekaran [69] stated that investment in training help organizations achieves a competitive advantage. Antony, McDermott, and Sony [37] argued that the organizational workforce requires new skills and training to achieve organizational readiness in I4.0. The growing body of literature recognizes the role of training and learning in achieving I4.0 readiness [8,43]. Khin and Kee [8] argued that the alignment of HRM with I4.0 is expected to enable teamwork and learning. Stentoft et al. [11] affirmed that strategy, customer requirement, data availability, and advanced technologies are the drivers of I4.0 readiness. The review study by Sony and Naik [54] stated that TMC, employee adaptability, strategy, IT-product and services, and extent of digitization are key ingredients for I4.0 readiness. They further argued that these factors are interrelated. Thus, an organization should consider these readiness factors in totality while implementing I4.0. Mittal, Khan, Romero, and Wuest [10] stated that strategy, leadership, operations, and technology are the essential factors for I4.0 readiness in manufacturing SMEs. The following hypotheses have been proposed based on the above literature and theoretical support. Figure 1 shows the research model.

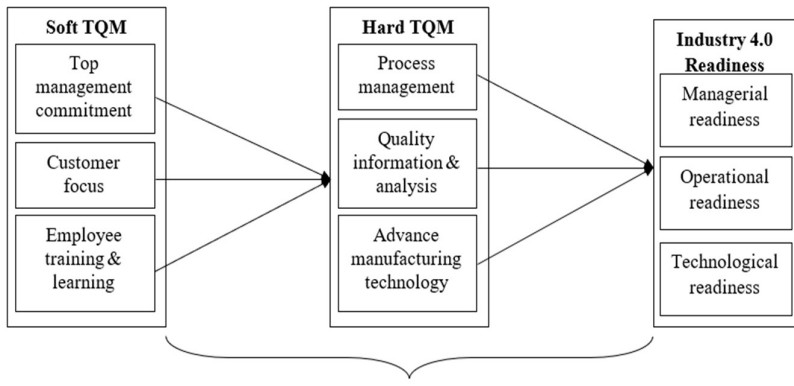

**Figure 1.** Research Model.

**H5a-c.** *PM positively and significantly affects I4.0 readiness (MR, OR, and TR).*

**H6a-c.** *QIA positively and significantly affects I4.0 readiness (MR, OR, and TR).*

**H7a-c.** *AMT positively and significantly affects I4.0 readiness (MR, OR, and TR).*

## 4. Material and Methods

The research methodology section describes the sampling process and data collection. Moreover, this section underscores the measurement of the constructs and statistical tests used to examine the effect of TQM 4.0 on I4.0 readiness.

### 4.1. Sampling and Data Collection

An online survey technique was used in the sampling process to reach the study population. Granello and Wheaton [70] argued that online survey research facilitates the researchers to reach unique populations and save time and cost. Likewise, Tanner [71] highlights the significant role of digital technology in survey techniques to reach the respondent with limited resources and time. The study population consists of small and medium manufacturing enterprises situated in Malaysia. The selection of SMEs was based on the number of full-time employees. According to SME Corporation Malaysia [72], a small firm has employees between 5–74, and a medium firm has employees between $75 \leq$ and 200. The data were collected from four states (Kelantan, Perak, Selangor, and Johr) of Malaysia. The justification for state selection is that these states grow faster than the national average in the manufacturing segment. Furthermore, the contribution of these states to the national GDP was higher (i.e., 41%) [73]. The simple random sampling technique was adopted to collect the data. Generally, the quality managers are well-versed in their organization's quality practices and are considered study respondents. Before conducting the survey, the questionnaire was pretested with industry experts in quality practices. The purpose of the pretesting was to guarantee that the prospective respondents could understand the questionnaire items. After pretesting, minor changes were applied to the questionnaire to obtain more clarity. Furthermore, ethical issues were considered during data collection.

To perform the actual survey, more than 750 emails were sent, and reminder emails were also sent. Based on the collected data, 229 survey questionnaires were collected with a response rate of 30.53%. After performing the data cleaning steps, 209 questionnaires were used for the final analysis. The data were analyzed using SmartPLS software version. It is a second-generation structural equation modeling software that can model latent variables with minimal requirements [74]. Therefore, this study employed Partial Least Square (PLS) path modeling analysis, a Structural Equation Modeling method (SEM). Based on the prior literature, SEM can analyze multiple relationships in one model [74]. Furthermore, PLS is preferred in this study due to its adaptability and less stringent assumptions. For example,

PLS is not based on the assumption of normality and is suitable for small samples, and it can be nominal, interval, or ratio [74]. Table 2 Show the demographic statistic of sample data.

**Table 2.** Demographic Statistics.

| Variable | Item | Frequency | Percentage (%) |
|---|---|---|---|
| Gender | Male | 127 | 60.77 |
| | Female | 82 | 39.23 |
| Firm size (employee) | Small (5–74) | 98 | 46.89 |
| | Medium (5–≤200) | 111 | 53.11 |
| Age of firm (Years) | Less than 5 years | 33 | 15.79 |
| | More than 5 but less than 10 | 89 | 42.59 |
| | More than 10 years | 87 | 41.62 |
| Industry type | Electrical and Electronics | 43 | 20.57 |
| | Chemical | 27 | 12.92 |
| | Textile | 31 | 14.83 |
| | Food | 59 | 28.23 |
| | Rubber and Plastic | 19 | 09.09 |
| | Machinery and Hardware | 13 | 06.22 |
| | Other | 17 | 08.14 |

*4.2. Measures*

The items of study variables are adapted from the past literature. Top management commitment (TMC) has been measured by four items adapted from Lin, et al. [75]. The six items of customer focus (CF) have been adapted from Jong, et al. [76]. The training and learning construct has been measured through three items adapted from Addis [77]. Process management (PM) has been measured through five items adapted from Abbas [78], and the quality information and analysis construct has been measured through eight items adapted from Sila [79]. Advanced manufacturing technology (AMT) has been measured through five items adapted from Iqbal, et al. [80]. Finally, the I4.0 readiness constructs are divided into managerial, operational, and technological. Six items of Managerial I4.0 readiness (MR), six items of operational I4.0 readiness, and four items of technological I4.0 readiness were adapted from Khin and Kee [8]. A seven-point Likert scale was used to measure the study items. The questionnaire is attached as Appendix A.

**5. Data Analysis**

*5.1. Descriptive Statistics and Common Method Bias*

This section highlights the mean, kurtosis, and skewness of study constructs. As recommended by Darren and Mallery [81], the descriptive analysis is performed through SPSS. Table 3 shows the descriptive analysis in detail. From the table, the mean value of all variables is above 4.0 except EDT (3.949) and QIA (3.646). Moreover, the skewness, kurtosis, and skewness values were within the threshold limit, i.e., ±2 [82].

Additionally, the common method bias (CMB) was performed to analyze the dataset's biasness. As Podsakoff [82] recommended, both procedural and statistical remedies were adopted to eliminate the CMB issue. From the statistical point of view, the Harman Single-factor test has been performed. The exploratory factor analysis highlights that a single factor accounted for less than 50% of the variance. In the context of PLS-SEM, Kock [83] and Kock and Lynn [84] argued that if the variance inflation factor (VIF) value through the full collinearity test is ≤3.3, then the dataset can be considered free from the CMB issue.

*5.2. Assessment of Measurement Model*

5.2.1. Reliability and Convergent Validity

As Hair, et al. [85] recommended, the measurement model assessment includes convergent validity. The first order constructs were used to test the reliability and validity [86]. In terms of reliability, Cronbach's Alpha, and rho_A were used to measure the data reliability.

As suggested by Hair, Page and Brunsveld [74], the Cronbach Alpha and rho_A value $\geq$ 0.70 are considered acceptable [87,88]. Regarding convergent validity, item loading and average variance extracted (AVE) are essential [74]. Hair, Sarstedt, and Ringle [85] suggest that the items loading $\geq$0.708 is deemed excellent. However, if the loading falls between 0.4–0.7 and AVE is $\geq$0.5, then researchers can be retained the constructs. Moreover, the threshold value of AVE is $\geq$0.5. Table 4 shows the reliability and convergent validity.

**Table 3.** Descriptive Analysis.

| Constructs | N | Mean | Kurtosis | Skewness |
|---|---|---|---|---|
| Top management commitment (TMC) | 209 | 4.031 | −1.285 | −0.022 |
| Customer focus (CF) | 209 | 4.600 | −1.083 | −0.343 |
| Employee training and learning (EDT) | 209 | 3.949 | −1.281 | 0.036 |
| Process management (PM) | 209 | 4.153 | −1.345 | −0.123 |
| Quality information and analysis (QIA) | 209 | 3.646 | −1.169 | 0.211 |
| Advance manufacturing technology (AMT) | 209 | 4.184 | −1.223 | −0.087 |
| Managerial I4.0 readiness | 209 | 4.086 | −1.297 | 0.023 |
| Operational I4.0 readiness | 209 | 4.435 | −1.217 | −0.188 |
| Technological I4.0 readiness | 209 | 4.034 | −1.057 | −0.021 |

**Table 4.** Constructs' Reliability and Convergent validity.

| Constructs | Items | Loadings (0.50–0.85) * | VIF (<5) ** | Reliability | | AVE (≥0.50) ** |
|---|---|---|---|---|---|---|
| | | | | Cronbach's Alpha (≥0.70) ** | rho_A (≥0.70) ** | |
| Top management commitment (TMC) | TMC1 | 0.884 | 2.631 | 0.901 | 0.902 | 0.770 |
| | TMC2 | 0.874 | 2.476 | | | |
| | TMC3 | 0.879 | 2.598 | | | |
| | TMC4 | 0.873 | 2.562 | | | |
| Customer focus (CF) | CF1 | 0.760 | 1.693 | 0.862 | 0.863 | 0.592 |
| | CF2 | 0.784 | 1.890 | | | |
| | CF3 | 0.758 | 1.778 | | | |
| | CF4 | 0.766 | 1.784 | | | |
| | CF5 | 0.762 | 1.788 | | | |
| | CF6 | 0.786 | 1.933 | | | |
| Training and learning (EDT) | EDT1 | 0.910 | 2.134 | 0.858 | 0.900 | 0.776 |
| | EDT2 | 0.879 | 2.302 | | | |
| | EDT3 | 0.852 | 2.058 | | | |
| Process management (PM) | PM1 | 0.856 | 2.536 | 0.912 | 0.915 | 0739 |
| | PM2 | 0.848 | 2.507 | | | |
| | PM3 | 0.853 | 2.420 | | | |
| | PM4 | 0.875 | 2.728 | | | |
| | PM5 | 0.867 | 2.602 | | | |
| Quality information and analysis (QIA) | QIA1 | 0.692 | 1.610 | 0.885 | 0.889 | 0.557 |
| | QIA2 | 0.699 | 1.605 | | | |
| | QIA3 | 0.758 | 1.862 | | | |
| | QIA4 | 0.732 | 1.721 | | | |
| | QIA5 | 0.723 | 1.717 | | | |
| | QIA6 | 0.742 | 1.756 | | | |
| | QIA7 | 0.767 | 1.923 | | | |
| | QIA8 | 0.846 | 2.499 | | | |

**Table 4.** *Cont.*

| Constructs | Items | Loadings (0.50–0.85) * | VIF (<5) ** | Reliability | | AVE (≥0.50) ** |
|---|---|---|---|---|---|---|
| | | | | Cronbach's Alpha (≥0.70) ** | rho_A (≥0.70) ** | |
| Advance manufacturing technology (AMT) | AMT1 | 0.766 | 1.841 | 0.862 | 0.869 | 0.644 |
| | AMT2 | 0.781 | 1.806 | | | |
| | AMT3 | 0.810 | 1.933 | | | |
| | AMT4 | 0.830 | 2.085 | | | |
| | AMT5 | 0.823 | 2.310 | | | |
| Managerial I4.0 readiness (MR) | MR1 | 0.845 | 2.567 | 0.927 | 0.930 | 0.731 |
| | MR2 | 0.862 | 2.706 | | | |
| | MR3 | 0.847 | 2.674 | | | |
| | MR4 | 0.868 | 2.874 | | | |
| | MR5 | 0.848 | 2.685 | | | |
| | MR6 | 0.860 | 2.673 | | | |
| Operational I4.0 readiness (OR) | OR1 | 0.900 | 3.809 | 0.955 | 0.956 | 0.818 |
| | OR2 | 0.898 | 3.724 | | | |
| | OR3 | 0.912 | 4.400 | | | |
| | OR4 | 0.915 | 4.510 | | | |
| | OR5 | 0.891 | 3.452 | | | |
| | OR6 | 0.909 | 4.093 | | | |
| Technological I4.0 readiness (TR) | TR1 | 0.803 | 1.684 | 0.796 | 0.811 | 0.622 |
| | TR2 | 0.831 | 1.825 | | | |
| | TR3 | 0.829 | 1.715 | | | |
| | TR4 | 0.682 | 1.352 | | | |

* [89] ** [90].

### 5.2.2. Discriminant Validity

In the PLS-SEM literature, three approaches are given to measure the discriminant validity: Fornell–Larker, cross-loading, and Heterotrait–Monotrait ratio (HTMT). Recently, Henseler, et al. [90] criticized that Fornell–Larker and cross-loading approaches cannot detect discriminant validity in PLS-SEM [91]. Therefore, the HTMT approach was adopted to perform the discriminant validity. The acceptable value of HTMT is ≤0.90 for similar variables or ≤0.85 for distinct variables [85]. Table 5 illustrate the discriminant validity (HTMT) of all latent variables, and all the constructs achieve the threshold limit (≤0.85).

**Table 5.** Discriminant Validity (HTMT).

| Latent Construct | (1) | (2) | (3) | (4) | (5) | (6) | (7) | (8) | (9) |
|---|---|---|---|---|---|---|---|---|---|
| TMC (1) | | | | | | | | | |
| CF (2) | 0.495 | | | | | | | | |
| EDT(3) | 0.449 | 0.494 | | | | | | | |
| PM(4) | 0.492 | 0.649 | 0.346 | | | | | | |
| QIA(5) | 0.782 | 0.498 | 0.428 | 0.511 | | | | | |
| AMT (6) | 0.581 | 0.355 | 0.153 | 0.275 | 0.556 | | | | |
| MR (7) | 0.409 | 0.658 | 0.515 | 0.424 | 0.512 | 0.358 | | | |
| OR (8) | 0.764 | 0.501 | 0.273 | 0.507 | 0.703 | 0.676 | 0.439 | | |
| TR (9) | 0.596 | 0.461 | 0.515 | 0.408 | 0.493 | 0.513 | 0.366 | 0.581 | |

### 5.3. Assessment of Structural Model

As recommended by Hair, Page, and Brunsveld [74], the association between exogenous and endogenous variables were examined in the structural model assessment. Table 6 shows the structural model results, effect size, and hypothesis accepted or rejected criteria. Based on the results, the association between soft TQM (TMC, CF, and EDT) and hard TQM (PM, QIA, and AMT) were examined. The association between TMC→PM (H1a: β = 0.243, $f^2$ = 0.072, $p < 0.05$), TMC→QIA (H1b: β = 0.610, $f^2$ = 0.581, $p < 0.05$), and TMC→AMT (H1c: β = 0.511, $f^2$ = 0.279, $p < 0.05$) are positive and significant. Hence, H1a, H1b, and H1c were accepted. Furthermore, the relationship between CF→PM (H2a: β = 0.467, $f^2$ = 0.258, $p < 0.05$), CF→QIA (H2b: β = 0.132, $f^2$ = 0.026, $p < 0.05$), and

CF→AMT (H2c: β = 0.149, $f^2$ = 0.023, $p < 0.05$) are positive and significant. Thus, H2a, H2b, and H2c were accepted. However, the relationship between EDT→PM (H3a: β = 0.010, $f^2$ = 0.001, $p > 0.05$), EDT→QIA (H3b: β = 0.082, $f^2$ = 0.011, $p > 0.05$), and EDT→AMT (H3c: β = −0.142, $f^2$ = 0.022, $p > 0.05$) were insignificant. Thus, H3a, H3b, and H3c are rejected. Based on the above, this highlights that soft TQM practices have significant and positive association with hard TQM practices except employee training and learning (EDT) has insignificant relationship with PM, QIA, and AMT.

**Table 6.** Structural Model and Effect Size.

| | Relation | β | *t*-Value | $f^2$ | CI [2.05%–97.5%] | Decision |
|---|---|---|---|---|---|---|
| H1a | TMC→PM | 0.243 | 3.464 | 0.072 | [0.084–0.368] | Accepted |
| H1b | TMC→QIA | 0.610 | 10.779 | 0.581 | [0.483–0.705] | Accepted |
| H1c | TMC→AMT | 0.511 | 7.291 | 0.279 | [0.363–0.631] | Accepted |
| H2a | CF→PM | 0.467 | 6.797 | 0.258 | [0.334–0.595] | Accepted |
| H2b | CF→QIA | 0.132 | 2.029 | 0.026 | [0.002–0.258] | Accepted |
| H2c | CF→AMT | 0.149 | 2.271 | 0.023 | [0.024–0.279] | Accepted |
| H3a | EDT→PM | 0.010 | 0.167 | 0.001 | [−0.100–0.138] | Rejected |
| H3b | EDT→QIA | 0.082 | 1.439 | 0.011 | [−0.016–0.197] | Rejected |
| H3c | EDT→AMT | −0.142 | 2.251 | 0.022 | [−0.268–0.019] | Rejected |
| H5a | PM→MR | 0.223 | 3.155 | 0.054 | [0.080–0.364] | Accepted |
| H5b | PM→OR | 0.215 | 4.508 | 0.085 | [0.107–0.300] | Accepted |
| H5c | PM→TR | 0.189 | 2.530 | 0.039 | [0.049–0.334] | Accepted |
| H6a | QIA→MR | 0.305 | 4.060 | 0.081 | [0.164–0.450] | Accepted |
| H6b | QIA→OR | 0.359 | 5.444 | 0.192 | [0.232–0.492] | Accepted |
| H6c | QIA→TR | 0.191 | 2.488 | 0.032 | [0.040–0343] | Accepted |
| H7a | AMT→MR | 0.121 | 1.610 | 0.015 | [−0.026–0.266] | Rejected |
| H7b | AMT→OR | 0.384 | 5.823 | 0.262 | [0.251–0.515] | Accepted |
| H7c | AMT→TR | 0.293 | 4.456 | 0.090 | [0.166–0.413] | Accepted |

Furthermore, Table 6 highlight the association between hard TQM practices and I4.0 readiness (MR, OP and TR). The results highlight that PM→MR (H5a: β = 0.223, $f^2$ = 0.054, $p < 0.05$), PM→OR (H5b: β = 0.215, $f^2$ = 0.085, $p < 0.05$), and PM→TR (H5c: β = 0.189, $f^2$ = 0.039, $p < 0.05$) were positive and significant. Hence, H5a, H5b, and H5c were accepted. Moreover, the results highlight that QIA→MR (H6a: β = 0.305, $f^2$ = 0.081, $p < 0.05$), QIA→OR (H6b: β = 0.359, $f^2$ = 0.192, $p < 0.05$), and QIA→TR (H6c: β = 0.191, $f^2$ = 0.032, $p < 0.05$) were positive and significant. Hence, H6a, H6b, and H6c were accepted. Finally, the results highlight that AMT→MR (H7a: β = 0.121, $f^2$ = 0.015, $p < 0.05$), AMT→OR (H7b: β = 0.384, $f^2$ = 0.262, $p < 0.05$), and AMT→TR (H7c: β = 0.293, $f^2$ = 0.090, $p < 0.05$) were positive and significant. Hence, H7a, H7b, and H7c were accepted.

*5.4. Mediation Analysis*

Table 7 summarizes the mediation effect of soft TQM (TMC, CF, and EDT) practices on I4.0 readiness (MR, OR, and TR) through hard TQM (PM, QIA, and AMT) practices. The PLS-SEM technique was adopted to perform the mediation analysis through SmartPLS. The results highlight that the Relationship between TMC→MR and OR, CF→MR, OR, and TR was mediated by hard TQM practices (PM). Whereas the Relationship between TMC→TR, EDT→MR, OR, and TR was not mediated by PM. Thus, PM partially mediated between soft TQM practices and I4.0 readiness, and H4a was partially supported. Furthermore, the results highlight that QIA is mediating between TMC→MR, OR, and TR. Surprisingly, QIA has not a significant mediator between CF→IR4.0, and EDT→IR4.0. Finally, Table 7 illustrates that AMT mediates between different sub-dimensions of soft TQM practices and IR4.0. The results show that the association between TMC→OR and TR, CF and AMT positively and significantly mediated OR and TR, and EDT (OR and TR). Thus, H4c is partially accepted. Figure 2 shows the output of SmartPLS.

**Table 7.** Mediation Analysis.

| | Relation | β | *t*-Value | *p*-Value | Decision |
|---|---|---|---|---|---|
| | TMC→PM→MR | 0.054 | 2.421 | 0.016 | Supported |
| | TMC→PM→OR | 0.052 | 2.407 | 0.016 | Supported |
| | TMC→PM→TR | 0.046 | 1.888 | 0.060 | Not Supported |
| | CF→PM→MR | 0.104 | 2.603 | 0.010 | Supported |
| H4a | CF→PM→OR | 0.101 | 3.789 | 0.000 | Supported |
| | CF→PM→TR | 0.088 | 2.358 | 0.019 | Supported |
| | EDT→PM→MR | 0.002 | 0.159 | 0.874 | Not Supported |
| | EDT→PM→OR | 0.002 | 0.164 | 0.870 | Not Supported |
| | EDT→PM→TR | 0.002 | 0.147 | 0.883 | Not Supported |
| | TMC→QIA→MR | 0.186 | 3.967 | 0.000 | Supported |
| | TMC→QIA→OR | 0.219 | 4.298 | 0.000 | Supported |
| | TMC→QIA→TR | 0.117 | 2.294 | 0.022 | Supported |
| | CF→QIA→MR | 0.040 | 1.723 | 0.086 | Not Supported |
| H4b | CF→QIA→OR | 0.047 | 1.872 | 0.062 | Not Supported |
| | CF→QIA→TR | 0.025 | 1.541 | 0.124 | Not Supported |
| | EDT→QIA→MR | 0.025 | 1.177 | 0.240 | Not Supported |
| | EDT→QIA→OR | 0.030 | 1.393 | 0.164 | Not Supported |
| | EDT→QIA→TR | 0.016 | 1.101 | 0.272 | Not Supported |
| | TMC→AMT→MR | 0.062 | 1.652 | 0.099 | Not Supported |
| | TMC→AMT→OR | 0.197 | 3.909 | 0.000 | Supported |
| | TMC→AMT→TR | 0.150 | 3.565 | 0.000 | Supported |
| | CF→AMT→MR | 0.018 | 1.060 | 0.290 | Not Supported |
| H4c | CF→AMT→OR | 0.057 | 2.143 | 0.033 | Supported |
| | CF→AMT→TR | 0.044 | 2.045 | 0.041 | Supported |
| | EDT→AMT→MR | −0.017 | 1.415 | 0.158 | Not Supported |
| | EDT→AMT→OR | −0.055 | 2.193 | 0.029 | Supported |
| | EDT→AMT→TR | −0.042 | 2.241 | 0.025 | Supported |

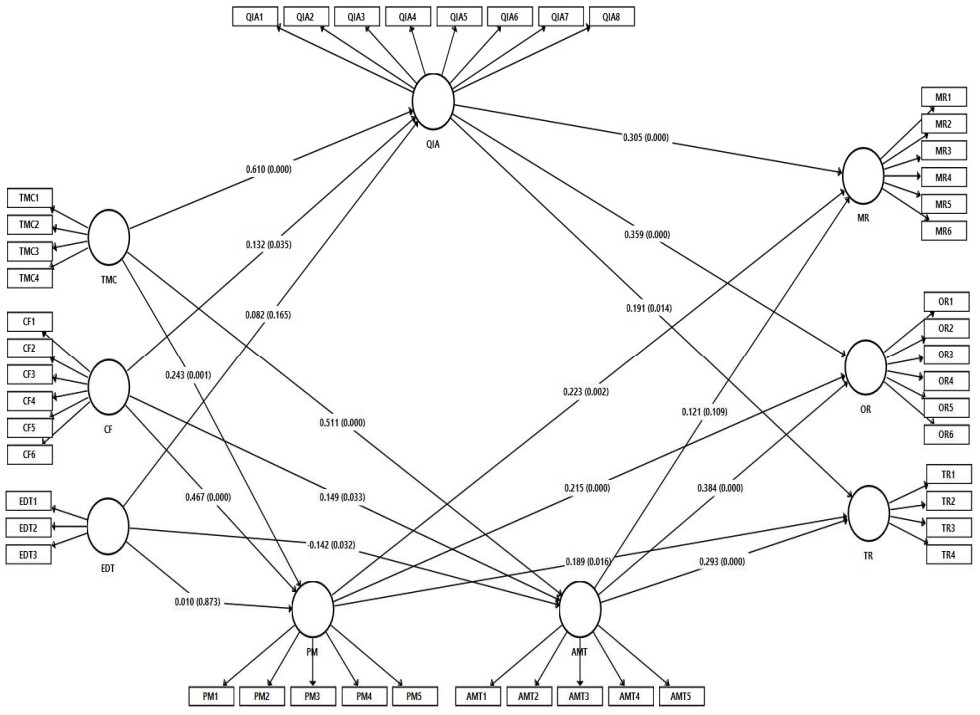

**Figure 2.** Results of Hypothesized Model.

*5.5. Artificial Neural Network (ANN) Analysis*

This section explains the logical reasons for applying artificial neural networks (ANN) and presents the results. Structural equation modeling (SEM) and multiple regression anal-

ysis (MRA) explain the linear relationship between exogenous and endogenous variables. However, both approaches are considered inadequate in explaining the complex nature of the decision-making process [92]. Furthermore, SEM and MRA are based on the compensatory assumption, which means that a decrease in one or more exogenous components in the framework can be compensated with the increase in other components [93]. However, in this study, the TQM practices are non-compensable and critical for IR4.0 [1,16,21,41]. For instance, a decrease in TMC cannot be compensated with an increase in CF and EDT as the exogenous constructs are distinctive in conceptualization and definitions. Hence, they are not interchangeable. ANN is performed with PLS-SEM to address this problem to capture the linear and non-linear relationships within a non-compensatory framework [94]. Furthermore, the integration of SEM and ANN provides more in-depth statistics that contribute to a more precise measurement of the association between each exogenous construct. Additionally, the application of two-stage SEM-ANN is a unique methodological contribution. Given the contributions and acceptability of the ANN technique, this study applied ANN to measure the association between each predictor (TMC, CF, EDT, QIA, PM, and AMT) and the dependent variable (MR, OR, and TR).

The architecture of the ANN technique consists of input, hidden, and output layers. Based on the prior literature, the feed-forward-back-propagation algorithm was used with multilayer perceptron [93–95]. Similar to Lim, Lee, Foo, Ooi and Tan [94], 90 percent of samples were allocated for training and 10 percent for testing. Wong, Leong, Hew, Tan, and Ooi [93] suggested that a ten-fold cross-validating approach was employed to minimize the over-fitting possibility and obtain the root mean square of errors (RMSE). In the ANN approach, sensitivity analysis is considered the essential part. It helps to analyze the predictive power of each input neuron. Table 8 shows the RMSE and sensitivity analysis between input (TQM practices) and output neuron (MR). The outcomes of sensitivity analysis demonstrate that CF has the greatest effect on MR with 100% normalized relative importance, followed by EDT (63%) and QIA (58%). Table 9 shows the RMSE and sensitivity analysis between TQM practices and OR. The outcomes of sensitivity analysis illustrate that AMT has the greatest effect on OR with 100% normalized relative importance, followed by QIA (94%), TMC (87%), and PM (71%). Finally, Table 10 shows the RMSE and sensitivity analysis between TQM practices and TR. The outcomes of sensitivity analysis indicate that TMC has the greatest effect on TR with 100% normalized relative importance, followed by EDT (98%), AMT (88%), QIA (78%), and PM (70%).

**Table 8.** RMSE and Sensitivity Analysis (MR as Dependent Variable).

| NN | Training RMSE | Testing RMSE | TMC | CF | EDT | QIA | PM | AMT |
|---|---|---|---|---|---|---|---|---|
| 1st | 0.535 | 0.647 | 0.056 | 0.342 | 0.215 | 0.236 | 0.074 | 0.078 |
| 2nd | 0.541 | 0.538 | 0.056 | 0.342 | 0.215 | 0.236 | 0.074 | 0.078 |
| 3rd | 0.567 | 0.512 | 0.052 | 0.387 | 0.280 | 0.149 | 0.122 | 0.010 |
| 4th | 0.567 | 0.579 | 0.132 | 0.353 | 0.275 | 0.059 | 0.091 | 0.089 |
| 5th | 0.561 | 0.519 | 0.139 | 0.360 | 0.131 | 0.247 | 0.035 | 0.088 |
| 6th | 0.564 | 0.495 | 0.011 | 0.320 | 0.189 | 0.363 | 0.108 | 0.009 |
| 7th | 0.544 | 0.561 | 0.034 | 0.328 | 0.237 | 0.222 | 0.042 | 0.137 |
| 8th | 0.558 | 0.634 | 0.021 | 0.300 | 0.320 | 0.218 | 0.041 | 0.100 |
| 9th | 0.563 | 0.514 | 0.063 | 0.425 | 0.198 | 0.134 | 0.076 | 0.104 |
| 10th | 0.530 | 0.543 | 0.062 | 0.437 | 0.195 | 0.198 | 0.059 | 0.049 |
| Mean | 0.553 | 0.554 | 0.183 | 0.982 | 0.619 | 0.571 | 0.216 | 0.192 |
| S.D | 0.014 | 0.052 | | | | | | |
| IMP. | | | 19% | 100% | 63% | 58% | 22% | 20% |

**Table 9.** RMSE and Sensitivity Analysis (OR as Dependent Variable).

| NN | Training | Testing | TMC | CF | EDT | QIA | PM | AMT |
| | RMSE | RMSE | | | | | | |
|---|---|---|---|---|---|---|---|---|
| 1st | 0.512 | 0.442 | 0.249 | 0.018 | 0.059 | 0.176 | 0.180 | 0.319 |
| 2nd | 0.497 | 0.471 | 0.207 | 0.040 | 0.008 | 0.256 | 0.220 | 0.269 |
| 3rd | 0.491 | 0.495 | 0.187 | 0.045 | 0.061 | 0.311 | 0.186 | 0.211 |
| 4th | 0.481 | 0.500 | 0.216 | 0.052 | 0.034 | 0.202 | 0.208 | 0.287 |
| 5th | 0.505 | 0.403 | 0.190 | 0.046 | 0.049 | 0.260 | 0.179 | 0.277 |
| 6th | 0.438 | 0.563 | 0.190 | 0.022 | 0.083 | 0.248 | 0.154 | 0.303 |
| 7th | 0.456 | 0.523 | 0.246 | 0.019 | 0.058 | 0.229 | 0.211 | 0.238 |
| 8th | 0.500 | 0.448 | 0.218 | 0.015 | 0.035 | 0.264 | 0.195 | 0.272 |
| 9th | 0.478 | 0.510 | 0.256 | 0.010 | 0.048 | 0.249 | 0.164 | 0.274 |
| 10th | 0.510 | 0.448 | 0.340 | 0.004 | 0.040 | 0.276 | 0.158 | 0.181 |
| Mean | 0.487 | 0.480 | 0.795 | 0.095 | 0.164 | 0.859 | 0.650 | 0.918 |
| S.D | 0.024 | 0.047 | | | | | | |
| IMP | | | 87% | 10% | 18% | 94% | 71% | 100% |

**Table 10.** RMSE and Sensitivity Analysis (TR as Dependent Variable).

| NN | Training | Testing | TMC | CF | EDT | QIA | PM | AMT |
| | RMSE | RMSE | | | | | | |
|---|---|---|---|---|---|---|---|---|
| 1st | 0.632 | 0.668 | 0.035 | 0.186 | 0.390 | 0.063 | 0.272 | 0.054 |
| 2nd | 0.613 | 0.572 | 0.464 | 0.073 | 0.148 | 0.098 | 0.137 | 0.080 |
| 3rd | 0.620 | 0.484 | 0.212 | 0.057 | 0.279 | 0.074 | 0.123 | 0.254 |
| 4th | 0.638 | 0.679 | 0.215 | 0.057 | 0.203 | 0.231 | 0.200 | 0.095 |
| 5th | 0.615 | 0.566 | 0.345 | 0.077 | 0.210 | 0.060 | 0.070 | 0.239 |
| 6th | 0.614 | 0.561 | 0.162 | 0.181 | 0.123 | 0.248 | 0.077 | 0.209 |
| 7th | 0.606 | 0.566 | 0.284 | 0.085 | 0.099 | 0.171 | 0.138 | 0.224 |
| 8th | 0.599 | 0.654 | 0.187 | 0.158 | 0.254 | 0.121 | 0.101 | 0.179 |
| 9th | 0.595 | 0.591 | 0.134 | 0.036 | 0.133 | 0.223 | 0.253 | 0.220 |
| 10th | 0.584 | 0.596 | 0.147 | 0.065 | 0.245 | 0.227 | 0.103 | 0.212 |
| Mean | 0.612 | 0.594 | 0.730 | 0.337 | 0.718 | 0.570 | 0.511 | 0.639 |
| S.D | 0.016 | 0.059 | | | | | | |
| IMP. | | | 100% | 46% | 98% | 78% | 70% | 88% |

## 6. Discussion, Implications, and Conclusion

### 6.1. Discussion of Findings

This study's prime objective is to examine the association between the multidimensional view of TQM and I4.0 readiness (MR, OR, and TR). Second, evaluate the importance of soft and hard TQM practices to achieve I4.0 readiness. Seven hypotheses were developed and empirically analyzed through PLS-SEM to accomplish the first objective. After that, the artificial neural network (ANN) approach was adopted to achieve the second objective. The following sections discuss the findings in more detail.

#### 6.1.1. Research Objectives-I

The first study objective examined the association between TQM practices (soft and hard) and I4.0 readiness (MR, OR, and TR). To achieve this objective, first, the authors examined the direct relationship between soft and hard TQM practices. Thus H1a-c, H2a-c, and H3a-c have been developed. Additionally, this study evaluates the direct association between TQM practices (hard) and I4.0 readiness (MR, OR, and TR); hence, H5a-c, H6a-c, and H7a-c have been developed. Finally, H4a-c was developed to examine the mediating role of hard TQM practices between soft TQM practices and I4.0 readiness.

Firstly, the empirical results highlight that soft TQM practices are positively and significantly associated with hard TQM practices. Generally, the study findings are consistent with past research studies [48,51,63,64,66]. Furthermore, the findings support the STS per-

spective that organizations must adopt social and technical systems before implementing new technology [54]. The past empirical studies confirmed that TMC has a positive and significant relationship with hard TQM practices [51,66,96]. Therefore, H1a, H1b, and H1c have been accepted. Lagrosen and Lagrosen [97] stated that the commitment of top management and all of its employees is critical for the QM's success. This commitment and involvement of top management and employees with outcomes must be embodied in the formulating and effectively implementing a set of strategies, actions, and policies related to the resources and processes (hard TQM). The prior literature affirmed that CF is positively and significantly associated with hard TQM practices [49,50,96]. Thus, H2a, H2b, and H2c have been accepted. Surprisingly, the empirical findings highlight that employee training and learning has no significant relationship with hard TQM practices. These findings are consistent with the past empirical study of Dow, et al. [98], Marri, et al. [99], and Santos, Sá, Félix, Barreto, Carvalho, Doiro, Zgodavová, and Stefanović [24]. The results of H3a-c have insignificant relationship with hard TQM practices. The results of this study are consistent with the past literature. For instance, Dow et al. [98] found that employee training programs have no relationship with AMT in Australian manufacturing firms. Similarly, Assen [100] affirmed that training aspects did not affect the usage of advance technologies in manufacturing organizations. Likewise, Santos et al. [23] highlighted that Portuguese quality managers have lack of digital skills to promote AMT in Quality 4.0. The SME Bank Malaysia [101] report highlighted that Malaysian SMEs faced the most significant challenge: the skill deficiencies among the workforces. Furthermore, the 12th Malaysian Plan (2021–2025) highlighted that the performance of Malaysian SMEs was low because of employees' lack of advanced skills and slow adoption of digital technology [13]. Thus, H3a, H3b, and H3c have been rejected.

Secondly, the findings confirmed that soft and hard TQM practices significantly affect I4.0 readiness. These findings are in line with the study conducted by Črešnar, Potočan, and Nedelko [17] and Sony et al. [16]. Because TQM is a multidimensional tool [102] and supports technology adaptation and utilization [50]. Furthermore, TQM tends to align with I4.0 principles and be considered appropriate solutions by organizations [17]. In addition, TQM digitalization provides a structure for achieving and integrating organizational quality practices through advances in automation, IT, and technology. Therefore, H5a-c, H6a-c, H7b, and H7c have been accepted. Unexpectedly, the findings highlight that AMT has no association with MR. Hence, H7a has been rejected.

Finally, the findings support the mediating effect of hard TQM practices in the relationship between soft TQM practices and I4.0 readiness. These findings support the notion that quality must come first as a prerequisite for other organizational outcomes Sciarelli, Gheith and Tani [51]. The results are also consistent with Zeng, Zhang, Matsui, and Zhao [48], who stated that improving quality would contribute to achieving other strategic goals over time.

### 6.1.2. Research Objective-II

The second research objective is to examine social and technical TQM factors' importance in promoting I4.0 readiness. To achieve this objective, an artificial neural network approach was adopted. The study results highlight that customer focus is vital to attain managerial I4.0 readiness in manufacturing SMEs, followed by employee education and training, quality information, and analysis. Maganga and Taifa [102] argued that customer-centeredness and knowledge and awareness are vital factors in achieving I4.0 readiness [95,103]. In the same vein, Maganga and Taifa [15] review study stated that training and big data are considered the enabling factors to adopt I4.0 technologies in manufacturing industries. The review study of Thekkoote [41] argued that data and analytics are considered vital for I4.0 adoption, followed by employee training. Additionally, the ANN results highlight that AMT is essential in achieving operational I4.0 readiness, followed by QIA, TMC, and PM. The empirical study of Nguyen, Tucek, and Pham [21] argued that TMC is considered the most critical indicator of I4.0 readiness. Sureshchandar [104] argued that leadership and quality and data management are the enabling factors

to achieve I4.0 readiness. Moreover, the results of ANN affirmed that TMC is considered vital to promote technology readiness among manufacturing SMEs, followed by EDT, AMT, QIA, and PM. The prior literature supports those social and technical TQM factors are essential to achieving I4.0 readiness [1,16,36,54,95].

## 7. Conclusions

### 7.1. Theoretical Contributions

From a theoretical point of view, firstly, this study contributes to the body of knowledge regarding the QM-I4.0 relationship by demonstrating the different effects of soft and hard TQM practices on I4.0 readiness in the SMEs sector. This allows for a greater degree of generalizability of results that have already been proven in larger organizations. Secondly, the multidimensional approach of TQM has proven essential and useful because different pathways lead through soft and hard practices that influence I4.0 readiness and practices. Furthermore, the findings also show how hard TQM is mediating. In other words, when processes and strategy are backed by good and committed leaders and effective human resources management, organizational performance is greatly enhanced. Thirdly, this research provides a reliable model based on empirical investigations that validate the theoretical relationships between the TQM practices and I4.0 readiness, which was previously only partially discussed in various studies with scant empirical evidence. Finally, the study shows that small and medium-sized enterprises have difficulty achieving the right technology readiness level because not all technologies are essential to all businesses.

### 7.2. Practical Implications and Conclusions

Overall, this research leads to a deeper understanding of the potential effects of soft and hard TQM practices on I4.0 readiness and actual practices of I4.0 in SMEs. Thus, it may serve as a guideline for SMEs. Based on the findings of this research, some implications are made for SME owners and managers. Firstly, the empirical evidence affirmed that soft TQM practices significantly affect hard TQM practices. This suggests that to implement I4.0 technologies properly, owners and managers should prioritize various soft practices relevant to management and employees. Secondly, the high importance of soft TQM on hard TQM emphasizes the interdependence of QM practices and the importance of using the STS approach to manage them. Therefore, managers must first set the foundations for quality by concentrating on soft TQM practices before implementing any quality improvement initiative. Thirdly, the low degree of association between readiness and actual implementation of I4.0 among SMEs can suggest an untapped potential for using I4.0 technologies to innovate business models. Despite the challenges of integrating I4.0 technologies, organizations should concentrate on their design strategies and technological integration. Lastly, practical implications also concern academia. In the sense of I4.0, previous studies have highlighted the value of digital education. As a result, academics should use these findings to update curricula and teaching methods and emphasize digital resources.

This study proposed and validated a conceptual framework based on STS theory which provides an integrating approach to combing soft and hard TQM practices to achieve I4.0 readiness (MR, OR, and TR). This study supports the notion that quality should be achieved first as a sequential precedent to other strategic capabilities. This study presents novel data for a research area lacking empirical data on soft and hard TQM and I4.0 readiness among SMEs. The empirical analysis supports the hypothesized relationships, except for the training and learning construct. Moreover, the results support that hard TQM practices mediate between soft TQM practices and I4.0 readiness. Furthermore, the ANN approach highlights that soft and hard TQM practices are essential to achieve I4.0 readiness in manufacturing firms. The findings are critical for businesses to consider as they prepare transition processes toward more digitalized processes.

### *7.3. Limitations and Future Research Directions*

Discussing the study findings, the study limitation, and future directions may be beneficial. Firstly, the analysis points out that there is still a gap between SMEs' I4.0 readiness and their I4.0 practices. Reactive technology investments partially cause this gap. Future studies could evaluate how various approaches—reactive and proactive—to adapting I4.0 technology are linked to the success of SMEs. Secondly, SME manufacturing was selected for this study. Therefore, the results cannot be generalized to other services industries. A comparative study can be conducted in the future to generalize the results better. Thirdly, the results are based on a single respondent's answers to a questionnaire survey. Building an analysis on a single person's point of view may be considered a limitation. Therefore, future research will investigate this phenomenon using a multi-respondent approach from each organization to improve the results. Finally, the potential effects of internal factors (organization culture, learning, strategic consensus) and external factors (regulatory conditions, technology turbulence, and competitive intensity) on the proposed framework may also be examined in future studies. These factors may be studied as moderators, generating more exciting results.

**Author Contributions:** Conceptualization, K.A. and S.K.J.; methodology, K.A. and R.F.A.; software, K.A., R.F.A. and A.A.; validation, S.K.J., A.M. and R.F.A.; formal analysis, K.A., A.A. and R.F.A.; investigation, S.K.J. and K.A.; resources, A.M., A.A. and R.F.A.; writing—original draft preparation, K.A. and S.K.J.; writing—review and editing, A.A., A.M. and R.F.A.; funding acquisition, R.F.A., A.M. and A.A. All authors have read and agreed to the published version of the manuscript.

**Funding:** This research was funded by Researchers Supporting Project, grant number RSP-2021/309, King Saud University, Riyadh Saudi Arabia.

**Institutional Review Board Statement:** Not applicable.

**Informed Consent Statement:** Not applicable.

**Data Availability Statement:** Available on request.

**Acknowledgments:** We are thankful for the help and guidance provided by all authors; without their assistance it would have been impossible to achieve such a research goal.

**Conflicts of Interest:** The authors declare no conflict of interest. The authors declare that they have no known competing financial interests or personal relationships that could have appeared to influence the work reported in this paper.

## Appendix A

| | |
|---|---|
| **Top Management Commitment (TMC)** | |
| Top management pay attention and actively discuss the quality technologies when adopting it. | |
| Top management provide highly support, such as HR and financial resources, to quality technologies. | Lin et al. [75] |
| Top management is willing to undertake the risk of implementing quality technologies. | |
| Top management encourage employees to apply digital quality in daily work. | |
| **Customer Focus (CF)** | |
| Our organization has been customer focused for a long time. | |
| Our organization provides mechanism for customer feedback. | |
| Our organization takes customer complaints as continuous improvement process. | Jong et al. [76] |
| Our organization reviews customer complaints and take into consideration for product innovation. | |
| Our organization conducts a customer satisfaction survey every year. | |
| Our organization conducts market study regularly to collect suggestions for improving our product. | |

| | |
|---|---|
| **Top Management Commitment (TMC)** | |
| **Employee training and learning (EDT)** | |
| Resources are available for digital quality related training in the company | Addis [77] |
| Training is given in the digital "Total quality and continuous improvement" concepts throughout the company. | |
| Training is given in the basic statistical techniques throughout the company. | |
| **Process management (PM)** | |
| Our organization has standardized operational processes which are clear and well understood by employees and customers. | Abbas [78] |
| Most of the processes in our organization are automated, fool-proof, and minimizes human error chances. | |
| Our organization has the latest technology and equipment to serve our customers more effectively and efficiently. | |
| Our system allows us to inspect and track key processes that are critical to the organization. | |
| Our organization regularly evaluates and improves their business process to ensure quality. | |
| **Quality information and analysis (QIA)** | |
| We collect and analyze organizational performance and cost data to identify and develop improvement. | Sila [79] |
| We examine customer-related/market data to develop priorities for improvement | |
| Our hardware and software systems are reliable and user friendly. | |
| We keep our information technology current with changing business needs and directions. | |
| We formally benchmark the best practices and performance of other industries. | |
| Quality data such as error and defect rate are available to managers and employees. | |
| We formally benchmark direct competitors product/services and processes. | |
| We use internet to provide high-quality data and information to employees, supplier, and customers. | |
| **Advance Manufacturing Technology (AMT)** | |
| Our organization uses Computer Aided Design (CAD) | Iqbal et al. [80] |
| Our organization uses Computer Aided Manufacturing (CAM) | |
| Our organization uses Flexible Manufacturing System (FMS) | |
| Our organization uses Robotics in production system. | |
| Our organization uses rapid prototyping for product development and design validation. | |
| **Managerial I4.0 Readiness** | |
| Our management is convinced that we should consider I4.0 production process. | Khin and Kee [8] |
| Our management has a plan to digitise the production process. | |
| Our management is mentally prepared to adopt I4.0. | |
| We have the right leadership in place to implement digitised production. | |
| Digital transformation is our corporate priority. | |
| Our management is convinced that we should consider I4.0 production process. | |
| **Operational I4.0 Readiness** | |
| Our company is financially prepared to digitalise production. | Khin and Kee [8] |
| Our staffs are cooperative in upgrading production processes. | |
| We are mentally prepared for changes in our production. | |
| We have staff to manage the I4.0 process. | |
| Our production machinery can be digitalised to I4.0. | |
| We have the infrastructure to support the I4.0 production process. | |

| Top Management Commitment (TMC) | |
|---|---|
| **Technological I4.0 Readiness** | |
| Our IT system could be upgraded for I4.0 production process. | Khin and Kee [8] |
| Our key machinery could be networked for I4.0 process. | |
| Our staffs are capable of learning new digital skills. | |
| Our staffs have sound knowledge about technical requirements for I4.0. | |

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
