# Peer review of "Soft and Hard Total Quality Management Practices Promote Industry 4.0 Readiness: A SEM-Neural Network Approach"

_sustainability, doi:10.3390/su141911917_

Round 1

Reviewer 1 Report

Dear Author,

1)  In Section 6.1.1, Furthermore, the findings support the STS perspective that organizations must adopt social and technical systems before implementing new technology (Sony & Naik, 2020). (Put in the bracket)

2) Add the latest references (2018-2022), especially for the introduction and literature review. 

3) Follow the guideline for references. 

Author Response

Point 1: In Section 6.1.1, Furthermore, the findings support the STS perspective that organizations must adopt social and technical systems before implementing new technology (Sony & Naik, 2020). (Put in the bracket).

Response 1: We want to thank the reviewer for this valuable comment, which stimulated us to make our paper clearer to the reader. We have updated the citation.

Point 2:  Add the latest references (2018-2022), especially for the introduction and literature review.

Response 2: Thanks for this valuable comment, in the introduction section all the references are from 2018 to 2022 (i.e., 4-refereces 2018, 6-references 2020, 8-references 2021, and 6-references 2022). Furthermore, the literature review also have latest references expect for basic variable definition like [35, 47, 49, 50, 57]. In a nutshell, the past references are not more than 10%.  

Reviewer 2 Report

Name of the Paper: Soft and Hard Total Quality Management Practices Promotes Industry 4.0 readiness: A SEM-Neural Network Approach

The present study empirically examines the effect of the multidimensional view of TQM (soft and hard) on Industry 4.0 readiness in small and medium-sized (SMEs) manufacturing firms using an online survey of 209 Malaysian SMEs manufacturing firms. The results show that the soft and hard TQM practices have supported Industry 4.0 readiness. Moreover, the results highlight that hard TQM practices have a mediating role between soft TQM practices and I4.0 readiness. The ANN results affirmed that multidimensional variables of TQM are important for the effective implementation of I4.0 in manufacturing firms.

General Observation:

1)       Please refer to the abstract, “The ANN results affirmed that multidimensional variables of TQM are important for the effective implementation of I4.0 in manufacturing firms.”. The pillars of Industry 4.0 are different. The authors need to clarify this claimed result.

2)      Please refer to “Many qualitative and review studies suggested that effective TQM implementation help firms improve I4.0 implementations [18].”

The authors provide only a single reference, that too is not related to implementations.

[18] Broday, E.E., 2022. The evolution of quality: From inspection to quality 4.0. International Journal of Quality and Service Sciences.

3)      Please refer to Table 1. Past Industry 4.0 models. The model names of the industry side must be cited correctly.

4)      Authors need to check the citation style, for instance “Similar to the study of Kashif Ali and Satirenjit Kaur Johl [53] and Ali and Johl [1]” Authors are the same but cited differently. Similarly, others in the manuscript.

5)      The following hypotheses need corrections: TCM should be TMC

H1a: TCM has a positive relationship with process management (PM).

H1b: TCM has a positive relationship with quality information & analysis (QIA).

H1c: TCM has a positive relationship with advanced manufacturing technology.

6)      Please refer to “Pham [21] argued that traditional TQM practices are based on standardization, while I4.0 focuses more on technical.” Which is inaccurate, rather it heavily focuses on interconnectivity, automation, machine learning, and real-time data.

7)      Please refer to Table 3. Descriptive Analysis, wherein Training & learning (EDT) should be Employee training & learning (EDT)

8)      Please refer to Figure 1. Research Model. The soft and hard skills considered for Industry 4.0 don’t represent actual Industry 4.0 enablers.

9)      Please refer to the following hypothesis results.

H3a: EDT has a positive relationship with PM.

H3b: EDT has a positive relationship with QIA.

H3c: EDT has a positive relationship with AMT.

The findings don’t support the positive relationship for the above hypotheses, which is unclear.

10)  The inference drawn for the above hypothesis states “Based on the above, this highlight that soft TQM practices have a significant and positive association with hard TQM practices except training & learning (EDT).” However, the hypothesis relates EDT with PM, QIA, and AMT

11)  A sample questionnaire may be appropriate to understand the construct and outcome, hence may be appended.

Author Response

Point 1: Please refer to the abstract, “The ANN results affirmed that multidimensional variables of TQM are important for the effective implementation of I4.0 in manufacturing firms.”. The pillars of Industry 4.0 are different. The authors need to clarify this claimed result.

Response 1: We want to thank the reviewer for this valuable comment, we updated the sentence in abstract pg#1, line # 23-26.

Point 2: Please refer to “Many qualitative and review studies suggested that effective TQM implementation help firms improve I4.0 implementations [18].”

The authors provide only a single reference, that too is not related to implementations.

[18] Broday, E.E., 2022. The evolution of quality: From inspection to quality 4.0. International Journal of Quality and Service Sciences.

Response 2: We want to thank the reviewer for this valuable comment, to justify our claim we added more citation on pg#2, line#77.

Point 3: Please refer to Table 1. Past Industry 4.0 models. The model names of the industry side must be cited correctly.

Response 3: We want to thank the reviewer for this valuable comment, in table 1, the industry reference has been updated.

Point 4: Authors need to check the citation style, for instance “Similar to the study of Kashif Ali and Satirenjit Kaur Johl [53] and Ali and Johl [1]” Authors are the same but cited differently. Similarly, others in the manuscript.

Response 4: We want to thank the reviewer for this valuable comment, the authors used Endnote as reference software. Due to software this error occurs. But I manually override the references and updated the citation on pg#5, line # 204.

Point 5: The following hypotheses need corrections: TCM should be TMC

H1a: TCM has a positive relationship with process management (PM).

H1b: TCM has a positive relationship with quality information & analysis (QIA).

H1c: TCM has a positive relationship with advanced manufacturing technology.

Response 5: We want to thank the reviewer for this valuable comment, the authors correct the errors.

Point 6: Please refer to “Pham [21] argued that traditional TQM practices are based on standardization, while I4.0 focuses more on technical.” Which is inaccurate, rather it heavily focuses on interconnectivity, automation, machine learning, and real-time data.

Response 6 : We want to thank the reviewer for this valuable comment. The reference is Nguyen et al 2021 [21] not the Pham [21]. Nguyen et al [21] study anchoring on STS theory to apply Delphi and AHP techniques to explore key factors and specific indicators of TQM 4.0.

Point 7: Please refer to Table 3. Descriptive Analysis, wherein Training & learning (EDT) should be Employee training & learning (EDT).

Response 7: We want to thank the reviewer for this valuable comment. Table 3 has been updated.

Point 8: Please refer to the following hypothesis results.

H3a: EDT has a positive relationship with PM.

H3b: EDT has a positive relationship with QIA.

H3c: EDT has a positive relationship with AMT.

The findings don’t support the positive relationship for the above hypotheses, which is unclear.

Response 8: We want to thank the reviewer for this valuable comment. To support the hypotheses outcomes references have been added on pg#16, line#550-560.

Point 9: The inference drawn for the above hypothesis states “Based on the above, this highlight that soft TQM practices have a significant and positive association with hard TQM practices except training & learning (EDT).” However, the hypothesis relates EDT with PM, QIA, and AMT.

Response 9: We want to thank the reviewer for this valuable comment. The sentence has been rephrased on pg#11, line#447-450.

Point 10: A sample questionnaire may be appropriate to understand the construct and outcome, hence may be appended.

Response 10: We want to thank the reviewer for this valuable comment. The sampled questionnaire has been appended.

Reviewer 3 Report

The main question addressed by the research is related to the examination of the effect of the multidimensional view of TQM on I4.0 readiness in small and medium-sized manufacturing firms and summarize the goals, approaches, and conclusions of the paper entitled Soft and Hard Total Quality Management Practices Promotes Industry 4.0 readiness: A SEM-Neural Network Approach. This subject is relevant and interesting.

The key messages that are conveyed to the reader highlights the fact that hard TQM practices mediate between soft TQM practices and I4.0 preparation. In this context the ANN results stated that the multidimensional variables of TQM are important for the effective implementation of I4.0 in manufacturing companies. The findings can help managers prioritize soft and hard quality practices of firms promoting I4.0 implementation, especially in emerging economies – and these findings are the strong point of the study.

The paper is well written, and the text is clear and easy to read. The conclusions are consistent with the evidence and arguments presented and they address the main question posed.

The Abstract highlight the important findings of the study.

The paper has a well-written introduction, which sets out the argument and summarizes recent research related to the topic.

The introduction establishes the originality of the research aims by demonstrating the need for investigations in the topic area.

The research carried oud involves sufficient use of statistical analysis, and samplings.

The tables and the figures aid to understand, but I suggest considering evaluating the trends observed and explain the significance of the results to wider understanding.

A detailed methodology is used, and the data analysis was done systematically.

The trends support the paper's discussion and conclusions.

In this study are sufficient data points to support the trends described by the author.

The data analysis and discussion sections described the results obtained.

The conclusions reflect the aims.

There are minor editing issues: e.g., lines 274, 275, 276, 417, 326, 552.

The references are adequate, but please check and unify the citation style according to the guidelines (e.g., 12, 13, 20, 25, 30-34, 43, 55, 68, 73-77, 83, 88, 90,101…).

Author Response

Point 1: There are minor editing issues: e.g., lines 274, 275, 276, 417, 326, 552.

Response 1: We want to thank the reviewer for this valuable comment, we thoroughly review the document and try to remove the editing issues.

Point 1: The references are adequate, but please check and unify the citation style according to the guidelines (e.g., 12, 13, 20, 25, 30-34, 43, 55, 68, 73-77, 83, 88, 90,101…).

Response 1: We want to thank the reviewer for this valuable comment, we thoroughly check the reference and citation. 

Round 2

Reviewer 2 Report

Authors have successfully modified the manuscript.

Reviewer 3 Report

This version looks better than the previous version.